# Multi-Objective Optimization Applied to the Design of Sustainable Pedestrian Bridges

**DOI:** 10.3390/ijerph20043190

**Published:** 2023-02-11

**Authors:** Fernando Luiz Tres Junior, Víctor Yepes, Guilherme Fleith de Medeiros, Moacir Kripka

**Affiliations:** 1Graduate Program in Civil and Environmental Engineering (PPGEng), University of Passo Fundo, Km 292, BR 285, Passo Fundo 99052-900, RS, Brazil; 2Institute of Concrete Science and Technology (ICITECH), Universitat Politècnica de València, 46022 Valencia, Spain

**Keywords:** multi-objective optimization, pedestrian bridge, sustainability, harmony search, carbon emissions

## Abstract

The demand for more sustainable structures has been shown as a growing tendency, and engineers can use optimization techniques to aid in the design and sizing stage, achieving solutions that minimize its cost and environmental and social impacts. In pedestrian bridges, which are subjected to human-induced vibrations, it is also important to ensure the users’ comfort, besides the security verifications. In this context, the objective of this paper is to perform a multi-objective optimization of a steel-concrete composite pedestrian bridge, minimizing cost, carbon dioxide emissions, and vertical acceleration caused by human walking. For this, the Multi-Objective Harmony Search (MOHS) was applied to obtain non-dominated solutions and compose a Pareto Front. Two scenarios were considered with different unit emissions obtained from a life cycle assessment in the literature. Results show that by increasing 15% the structure cost, the vertical acceleration is reduced from 2.5 to 1.0 m/s^2^. For both scenarios, the optimal ratio for the web height and total span (*L_e_*) lies between *L_e_*/20 and *L_e_*/16. The web height, the concrete strength, and the slab thickness were the design variables with more influence on the value of the vertical acceleration. The Pareto-optimal solutions were considerably sensitive to the parameters varied in each scenario, changing concrete consumption and dimensions of the welded steel I-beam, evidencing the importance of carrying out a sensitivity analysis in optimization problems.

## 1. Introduction

The civil construction industry plays an important role in the economic and social development of countries, while at the same time resulting in major environmental impacts. The sector was responsible for 34% of energy consumption and 37% of total carbon dioxide emissions worldwide in 2021 [1]. Therefore, the search for sustainable solutions in this sector has been growing considerably, as can be seen by the number of items that civil construction is involved in the sustainable development goals of the United Nations [2]. Bigger constructions, such as bridges, also have bigger scales of impact, hence the special attention these types of structures receive in the literature [3].

For bridges, the sustainable performance of the structure is directly related to the initial stages of the design, starting from the choice of which material will be used [4]. In the research of Milani, Yepes, and Kripka [5], steel-concrete composite bridges are identified as the most suitable solution in terms of costs, lifespan, environmental impact, architecture, and security sensation. The sizing phase also represents a major influence on the sustainability of the structure. However, this process is traditionally performed by a trial-and-error method, highly influenced by engineers’ previous experiences and with no guarantee that the solution adopted is the best possible. In this situation, optimization techniques can be useful tools for aiding in the decision-making process. Several studies were developed on this theme. Pedro et al. [6] minimized the cost of bridges in steel-concrete composite beams. Kaveh and Zarandi [7] used three optimization techniques to reduce the costs in the design of steel-concrete composite bridges. Orcesi, Cremona, and Ta [8] performed a multiobjective optimization, considering the live cycle of the bridge. Montoya, Hernández, and Kareem [9] applied an aero-structural optimization to obtain the optimal shape and size of a bridge deck. Yepes et al. [10] developed a program for the optimization of steel-concrete composite pedestrian bridges.

In the case of pedestrian bridges, besides considering the cost and environmental impacts, pedestrian comfort can be one of the objectives of the optimization problem. With the growth in the slenderness of this type of structure, they became more susceptible to human-induced vibrations [11,12,13]. The accelerations generated by the vibrations can be uncomfortable for pedestrians and may cause structural damage in extreme cases. Including criteria for limiting and controlling the accelerations originated by the dynamic load can improve the comfort for those who use the structure.

Among the optimization techniques applied to structural engineering, the Harmony Search is a metaheuristic inspired by jazz musical improvisation, and its algorithm is based on memorization and improvisation. Although the original method was designed for mono-objective optimization, some publications successfully implement a multiobjective harmony search (MOHS). As examples, the research of Kougias and Theodossiou [14], Jeddi and Vahidinasab [15], Geem [16], and Alkhadashi [17] can be cited. The MOHS was also applied in the optimization of a post-tensioned concrete bridge in the publication of García-Segura and Yepes [18]. In multiobjective optimization, instead of obtaining a single optimal solution, the result is a set of non-dominated solutions, i.e., solutions whose objective values cannot be improved without trading off the worsening of another objective. This set forms the Pareto front, which can be used as an aid in decision-making. 

Researchers have applied optimization techniques in the design of pedestrian bridges with distinct objectives. Ferenc and Mikulski [19] performed a parametric optimization of a glass fiber-reinforced polymer aiming to minimize its self-weight. Ferreira and Simões [20] minimized the cost of a cable-stayed pedestrian bridge with control devices. Penadés-Plà, García-Segura, and Yepes [21] used a metamodel-aided optimization to minimize the embodied energy of a concrete box-girder pedestrian bridge. In another research, Ferreira and Simões [22] optimized a cable-stayed steel pedestrian bridge with viscous dampers with the objective of reducing costs while satisfying dynamic verifications. The sustainability of a concrete box-girder pedestrian bridge is optimized in the publication of García-Segura et al. [23], minimizing the cost and CO_2_ emissions. The optimization considering tuned mass dampers in pedestrian bridges is a topic of interest in different publications [24,25,26]. Although several studies have been developed regarding the optimization of costs and environmental impacts or the dynamic response of the pedestrian bridge, there is a lack of publications that consider the three objectives simultaneously. This justifies the research on the assessment of the interaction between the sustainability of the pedestrian bridge and its vertical acceleration generated by human-induced vibrations in a multiobjective optimization problem. Furthermore, the topic of optimization of steel-concrete composite pedestrian bridges is barely explored, as also pointed out by Yepes et al. [10].

In this research, in addition to focusing on the sustainable design of pedestrian bridges, pedestrian comfort is jointly evaluated in terms of vertical accelerations caused by human-induced vibrations. The objective of this paper is to provide subsidies for decision-making in the sustainable design of a pedestrian bridge with a steel-concrete composite structure, aiming to minimize the vertical accelerations, as well as the cost and the emission of carbon dioxide. For that, a Python program is developed, implementing an algorithm of the multiobjective harmony search and verification routines regarding the structure’s ultimate and serviceability limit states. Two scenarios for the CO_2_ unit emission of the concrete slab are studied, with different values obtained from a life cycle assessment presented by Santoro and Kripka [27]. This sensitivity analysis aims to verify the influence of this parameter in the Pareto-optimal solutions.

## 2. Materials and Methods

### 2.1. Multiobjective Harmony Search

Among the various optimization methods applied to structural engineering, the Harmony Search (HS) is an algorithm that has been showing competitive results. Proposed originally in 2001 by Geem, Kim and Loganathan [28], the HS was conceived inspired by jazz musical improvisation, where musicians search for the perfect harmony between the notes of each instrument. In this analogy, each instrument represents a variable of the optimization problem, and its notes are the assigned values that combine into a harmony, i.e., a solution. These harmonies are evaluated by an aesthetic estimation based on an objective function. Throughout the improvisations or iterations, the worst harmonies are discarded and replaced by the best ones, seeking to reach the global optimum of the problem.

Although the HS was designed to solve mono-objective optimization problems [29,30,31], there are publications proposing and implementing algorithms for a Multiobjective Harmony Search (MOHS), with a growing interest in scientific research [32]. Two formal proposals are presented by Ricart et al. [33], MOHS1 and MOHS2, which maintain the main characteristics of the original method, but change the ranking of solutions in the harmony memory. While the MOHS1 keeps the improvisation of a single solution in each iteration, the MOHS2 improvises a new harmony memory and then performs the ranking of the solutions.

From MOHS2, Sivasubramani and Swarup [34] published a similar algorithm, implementing a dynamic variation in the values of the pitch adjustment rate (PAR) and the bandwidth (bw) based on a well-known and accepted modification of the mono-objective HS [35]. It is also proposed to consider the crowding distance in the ranking step using the algorithm presented by Deb et al. [36]. This metric is used to increase the diversity and avoid agglomeration of solutions, providing a Pareto front with better conformation. Given these improvements, this is the algorithm implemented in this paper.

The algorithm of the MOHS used in this study can be described by the following 5 main steps, which are also illustrated in Figure 1.

Step 1. Initialize the objective functions, constraints, and parameters of the method: harmony memory size (HMS), harmony memory considering rate (HMCR), maximum and minimum pitch adjustment rate (PAR_min_ and PAR_max_), maximum and minimum bandwidth (bw_min_ and bw_max_) and the maximum improvisation number (MI).

Step 2. Harmony memory (HM) initialization with random starting solutions generated within the discrete search space.

Step 3. Improvise a new harmony memory (HM_2_) with the same number of harmonies as the HMS value. Each harmony can be generated in three different ways: random selection, memory consideration, or pitch adjustment. If a random number *rand_1_* is equal to or bigger than HMCR, the value of the variable is obtained by random generation; otherwise, it is obtained by memory consideration. In this case, a new random number *rand_2_* is compared to PAR, and if the *rand_2_* value is equal or bigger, the value of the variable is randomly selected from harmony in the HM. If *rand_2_* is smaller than PAR, then the new value is submitted to pitch adjustment, considering a value of the variable stored in HM and performing a step based on the bw parameter.

Step 4. Harmony memory update, starting with the union of HM and HM_2_ in a matrix H_u_, with 2*HMS harmonies. The solutions are ranked by a non-dominated sorting method proposed by Fonseca and Fleming [37]. The ranking value of each solution is equal to the number of solutions that dominate the solution evaluated plus one, so non-dominated solutions have a rank equal to 1. Then, the solutions are transferred to HM by rank order until the next rank has more solutions than the left space in HM. At this stage, the crowding distance of the solutions is evaluated using the algorithm presented by Deb et al. [36]. The harmonies with the greatest crowding distance are moved to HM until it is filled. 

Step 5. Stopping criterion check. If the maximum number of improvisations is achieved, the procedure is ended, and the HM is returned, containing the Pareto-optimal solutions that compose the Pareto Front. Otherwise, repeat steps 3, 4, and 5.

Table 1 summarizes the values adopted for each parameter of the MOHS algorithm implemented in this paper. Note that the HMS, in the multiobjective optimization, also means the number of solutions that will compose the Pareto-optimal set of solutions.

### 2.2. Problem Formulation and Implementation

The geometry of the pedestrian bridge consists of a steel-concrete composite structure with two welded I-beams, a concrete slab, and headed stud shear connectors to ensure the interaction between both materials. The choice of using two steel beams is based on previous research that points to this configuration as the lowest environmental impact solution for bridge-type structures [4]. It is considered a steel ASTM A572 grade 50 for the I-beams, with 350 MPa of yield tensile strength and 200 GPa of modulus of elasticity. The dimensions of the headed welding stud adopted are a nominal diameter of 19 mm and a height of 135 mm. The pedestrian bridge is simply supported, with a single span of 17.5 m to be surpassed. During the construction phase, it was considered that the steel beams were unpropped. As illustrated in Figure 2, the total width of the structure is 3 m, with the beams spaced by 1.8 m and external cantilevers 0.6 m in length. In the same figure, *d* is the total depth of the welded I-beams sections, and *h_t_* is the concrete slab thickness, both defined by the optimization process. The longitudinal section, as well as the span of the structure, are illustrated in Figure 3.

#### 2.2.1. Design Variables

The optimization problem variables (*x_i_*) are the characteristic strength of concrete (*f_ck_*), the concrete slab thickness, the steel beam dimensions, and the interaction degree (*α*) of the composite beam, as shown in Figure 4. In the illustration, *h_w_* is the web height, *t_w_*, *t_fs,_* and *t_fi_* are, respectively, the thickness of the web, of the superior and inferior flanges, and *b_fs_* and *b_fi_* are the width of the superior and inferior flanges. In total, 9 design variables are considered, and their values are discrete to ensure that the solutions are feasible and represent a real design. The possible intervals for each variable are shown in the following equations, which form the lateral constraints of the optimization problem. These discrete intervals also limit the minimum and maximum dimensions of the structural elements, as well as consider the manufacturing standard of Brazilian metallurgists.
(1)$$x_{1}\in \;\left[25;\;30;\;35;\;40;\;45;\;50\right]\;\mathrm{in}\;\mathrm{MPa}$$
(2)$$x_{2}\in \;\left[\;100;\;110;\;\cdots ;\;190;\;200\right]\;\mathrm{in}\;\mathrm{mm}$$
(3)$$x_{3}\in \;\left[150;160;\;\cdots ;\;1990;\;2000\right]\;\mathrm{in}\;\mathrm{mm}$$
(4)$$x_{4},\;x_{6},\;x_{8}\in \left[4.75;\;6.35;\;8.0;\;9.5;\;12.5;\;16.0;\;19.0;\;22.4;\;25.0;\;28.5;\;31.5;\;37.5;\;44.5;\;50.0\right]\;\mathrm{in}\;\mathrm{mm}$$
(5)$$x_{5},\;x_{7}\in \;\left[100;\;110;\;\cdots ;\;1990;\;2000\right]\;\mathrm{in}\;\mathrm{mm},$$
(6)$$x_{9}\in \;\left[0.4;\;0.5;\;0.6;\;0.7;\;0.8;\;0.9;\;1.0\right].$$

#### 2.2.2. Objective Functions and Emissions Scenarios

The proposed optimization problem aims to minimize both the cost and the CO_2_ emissions associated with the construction phase of the pedestrian bridge. Equation (7) is used to evaluate the monetary cost of the structure (*C*), in the function of the unit costs (*c_i_*) and the respective quantity of material consumed (*q_i_*), the elements (ec) considered and their unit is the volume of concrete, the mass of the slab’s reinforcement steel and the steel beam mass, which also considers the mass of the headed studs. Unit costs were obtained from the SINAPI report [38], made available by the Brazilian Federal Savings Bank. These values are given in Table 2, where the monetary costs are expressed in real (R$), the Brazilian currency. As a reference, on 22 January 2023, the exchange rate against the US dollar is R$ 1.00 for $ 0.19 (or $ 1.00 for R$ 5.21).
(7)$$inimize\;C\left(x\right)=\overset{ec}{\underset{i=1}{\sum }}c_{i}\cdot q_{i}\left(x\right).$$

The environmental impact of the structure construction is measured in terms of CO_2_ emissions (*E*) and evaluated similarly to the cost, as can be seen in Equation (8). The elements (*ee*) are the same as in the cost evaluation but consider a unit emission of CO_2_ (*e_i_*). For the steel I-beam, the value of unit emission adopted was provided by Worldsteel [39]. On the other hand, the unit emissions of each concrete strength and the steel reinforcement rebars are those published by Santoro and Kripka [27] and are summarized in Table 2.
(8)$$minimize\;E\left(x\right)=\overset{ee}{\underset{i=1}{\sum }}e_{i}\cdot q_{i}\left(x\right).$$

This paper proposes two optimization scenarios, which consider different unit emission values for the concrete and its reinforcement, based on results obtained by Santoro and Kripka [18]. In the research, the authors performed a cradle-to-gate Life Cycle Assessment (LCA) of concretes with different strengths, reinforcements, and formworks in a city in the south of Brazil. As a result, the authors quantify the CO_2_ emissions, considering the acquisition of raw materials, transport to the concrete plant, production in the concrete batch, and the concrete transport to the building site. More details are presented in the original publication.

The scenarios studied in this research are:Scenario (A)—unit emissions obtained from the on-site survey for the study region. In Table 2, emission A shows the values considered in this scenario.Scenario (B)—unit emissions calculated using the SimaPro software, with Ecoinvent 3.5 database and ReCiPe 2016 method, with adjustments in the processes and quantities to make the values compatible with the same region. These values are displayed as emission B in Table 2.

Finally, the third objective function is to minimize the vertical accelerations generated by human-induced vibrations, according to Equation (9). With this objective, it is sought to increase the comfort of the pedestrians using the structure. The formulation is provided by the Brazilian code ABNT NBR 7187 [40], where *K_a,_*_95*%*_, *C*, *K*_1_, *K*_2_ e *K_f_* are tabled constant values from the same code, *d* is the pedestrian density (with a value adopted of 1.0 pedestrian/m^2^), *L* is the structure length, *b* is the width of the pedestrian bridge, and *M_i_* is the modal mass, associated to mode *i*.
(9)$$minimize\;a_{max}\left(x\right)=K_{a,95\%}\cdot \frac{d\cdot L\cdot b}{M_{i}}\cdot \sqrt{C\cdot K_{f}^{2}\cdot K_{1}\cdot \varepsilon^{K_{2}}}.$$

#### 2.2.3. Verifications and Constraints

The stresses, displacements, and dynamic parameters of the structural model are analytically calculated through the implementation of the formulas in the Python program developed in this study. The analysis considered a linear behavior of the steel I-beams and concrete slab. As permanent loads, it was taken into account the self-weight of the structure and shear connectors (automatically calculated for each solution), a concrete regularization (0.3 kN/m^2^), and railing (1 kN/m). As variable actions, there is the constructive live load (1 kN/m^2^), the pedestrian live load (5 kN/m^2^) [41], as well as horizontal actions, such as wind [42] and a 100 kN punctual load in the most unfavorable situation to the structure.

The verifications of the ultimate limit state refer to the shear stress and the bending moment acting on the composite beam [43], as well as the bending moment on the concrete slab [44]. The steel beams are also verified for the bending moments acting before the concrete hardening, and because the I section beams are unpropped during the structure construction, the lateral buckling in the construction phase is verified as well [43], considering a bracing length of 2.5 m. It is also considered requirements regarding constructability and the geometry [43,45] of the welded steel beam, such as I section slenderness and ratios between depth, width, and web thickness. The depth of the beam must be 50% bigger than the inferior flange width, and the thickness of the flanges must be bigger than the web thickness. The interaction degree should meet a minimum value, calculated in function of the span and the ratio between the superior and inferior flanges areas, and is always bigger than 40%. The serviceability limit state is also verified, restricting displacements to 1/350 of the span. In addition to that, the natural frequency of the structure must be outside of the interval from 1.0 to 2.6 Hz, and the vertical accelerations generated by pedestrians should be below 2.5 m/s^2^ [41]. 

All the verifications above must be met for the solution to be considered feasible. In the optimization problem, this is handled by constraints in the form of normalized inequations created from the verifications. Solutions that fail any of the constraints are penalized by an increase in the objective function (in the case of minimization), turning the solution less efficient for the optimization algorithm. In this paper, the penalization is obtained by the sum of the products of the constraint by a penalty factor. This value was determined based on a series of previous tests, where the lowest penalty factor that resulted in feasible solutions after the optimization end was adopted, with a value of 10^4^ in this research.

## 3. Results and Discussion

The Pareto-optimal solutions for each scenario are presented in Figure 5, arranged by vertical acceleration versus cost and emission. These are the solutions whose value of an objective cannot be improved without worsening at least one of the other objectives. In both scenarios, the costs of non-dominated solutions are very similar and stay within the range of R$ 1400.00 ($ 268.82) and R$ 1600.00 ($ 307.23) per meter of the pedestrian bridge, with a singular exception that costs approximately R$ 2000.00 ($ 384.03) per meter. As expected, the cost is inversely proportional to the vertical acceleration, given the need to increase the structural stiffness to reduce the accelerations generated by vibration, which increases material consumption. However, a slight increase in the cost can provide a significant improvement in the comfort for pedestrians, e.g., to reduce the acceleration from 2.5 to 1.0 m/s^2^, it is necessary to invest R$ 200.00 ($ 38.40) per meter of the structure, which corresponds to an increase of 15% in the cost. To reduce the acceleration from 2.5 to 1.5 m/s^2^, it is necessary an increase of only 7% in the structure cost.

Optimization results for emission and cost are shown in Figure 6, with the Pareto-optimal solutions from scenarios (A) and (B). As expected, the Pareto front correspondent to scenario (A) has, in general, solutions with lower CO_2_ emissions due to the unit emissions being considerably lower in comparison to Scenario (B). It is also possible to notice that for Scenario (A), solutions are distributed almost linearly, with the emissions growing as the cost increases. On the other hand, in Scenario (B), the solutions are more dispersed, without a clear tendency between the two objectives. However, the cost and the CO_2_ emissions of the pedestrian bridge are not conflicting objectives, with solutions that are at the same time efficient in terms of costs and environmental impacts.

This behavior for Scenario (B) can be mainly explained by a difference in the Pareto-optimal cross-section configuration. Some solutions showed a preference for bigger web heights and smaller slab thickness and concrete strength, being the ones with the best performance in terms of pedestrian comfort and CO_2_ emissions. In contrast, other solutions adopted smaller web heights, bigger slab thickness, and higher concrete strengths, reducing the costs but trading off a poorer performance in terms of environmental impacts.

Still, this can be evidence that the reduction in the cost of the structure is also an effective way to achieve environmentally cleaner designs in terms of CO_2_ emissions. A possible explanation for this is that to minimize costs, the optimization algorithm favors solutions that reduce material consumption, which also reduces the environmental impact of the structure. Therefore, minimizing costs and emissions are not conflicting objectives, as the literature also points out [10,46,47], for different types of structures and materials.

The web height (*h_w_*) is a design variable that notoriously influences the cost, the environmental impact, and the vertical acceleration generated by human-induced vibrations. This occurs due to the influence of the variable in the structure’s resistance and stiffness. For Scenario (A), Pareto-optimal solutions with smaller *h_w_* have lower cost and environmental impact but perform poorer in comfort for the pedestrians, as Figure 7 shows. For the problem studied in this paper, the optimal values to *h_w_*, for Scenario (A) are in the range of 900 mm and 1100 mm. The figure also shows that increasing this variable is an effective way to reduce the vertical accelerations from 2.5 m/s^2^ to 1.0 m/s^2^. However, to reduce accelerations further, other variables should be considered, such as the superior and inferior flange width, slab thickness, and concrete strength. The change in the behavior occurs as an attempt to satisfy the constraints regarding slenderness and the bending moment about the weak axis of the I-beam.

Again, Scenario (B) presented a quite different behavior, although the interval of optimal values for *h_w_* is very similar. Two groups of Pareto-optimal solutions stand out in Figure 7. The first remains practically constant around the value of 850 mm, while the second is grouped near the value of 1050 mm. For this scenario, variables such as slab thickness, concrete strength, and superior and inferior flange width played a bigger role in obtaining solutions with lower vertical accelerations. The solutions with *h_w_* close to 850 mm have, in general, a slab thickness bigger than 14 cm and a preference for concrete strength of 50 MPa. The other configurations, with bigger *h_w_*, have the minimum slab thickness and concrete strength. For both cases, the optimal ratio for *h_w_* and total span (*L_e_*) lies between *L_e_*/20 and *L_e_*/16.

Regarding the constraints of the optimization problem, the ones that govern the structure sizing are, in the vast majority, the displacement, the I-beam slenderness, and the horizontal bending moment that acts in the weak axis of the steel beam. As displacement is the active restriction in almost every solution, the algorithm shows a preference for adopting the biggest *h_w_* that complies with the maximum slenderness, given that this is an effective way to increase the beam’s stiffness. However, solutions that increase only the slenderness ratio can have a lower resisting capacity in the weak axis of the I-beam profile, and because of that, the constraint related to the verification of horizontal loads is also important in the sizing of the beam. For this restriction to be satisfied, solutions must also increase the flanges width and thickness of the I beam cross-section.

In cost terms, the welded I-beam is the element with the vast majority contribution to the value, as shown in Figure 8, for the lowest cost solution. On the other hand, the concrete slab shares a considerable part of the structure’s emissions, as illustrated in Figure 9, for the lowest emission configuration. While the slab represents less than 5% of the total cost of the structure, 26% of the CO_2_ released in its construction is due to the structural element. In Scenario (B), the concrete slab contribution of emissions increases to almost 46%. For this reason, the results show a certain preference for solutions with the minimum slab thickness, reducing the volume of concrete consumed.

Varying the unit emissions of the concrete and its reinforcement caused changes in the optimal values for the design variables. The average value of each variable is presented in Figure 10. In scenario (B) optimization, there is a preference for higher resistance concretes and bigger slab thickness, while the opposite happens in scenario (A). Concerning the I-beam, the web height and thickness are bigger in scenario (A) optimization, while the inferior flange thickness is bigger in scenario (B). This increase in steel consumption can be explained as an attempt to compensate for the low consumption of concrete. In both cases, the complete interaction is the optimal degree in all solutions. 

As can be noticed from the above, the values adopted for unit emissions can significantly change the results, both in terms of quantifying the environmental impact and in the optimal configuration of the solutions. This shows the importance of ensuring the reliability of the LCA study so that its practical applications are valid and represent reality as faithfully as possible. Regarding the impacts on the optimization results, it becomes clear the need to carry out sensitivity analyses to better understand the behavior of the results under possible changes in the parameters of the problem. It is also important to emphasize that the optimal solutions are intrinsically linked to the problem modeling and its limitations, and special care must be taken when extrapolating such results to different situations and applications. 

Finally, the program developed in this research proved to be capable of obtaining a Pareto front of non-dominated solutions, which can aid in the decision-making process during the design of a steel-concrete composite pedestrian bridge. In addition, the MOHS algorithm proved to be applicable in the optimization of three distinct objectives for the structural engineering problem studied. It is worth mentioning that both the methodology of this work and the developed program are adaptable for different types of structures, given that certain changes in the code are made, such as in the structural analysis and the pertinent verifications. This allows the study of other structural problems focused on their sustainability, and the efficiency of the optimization algorithm in different situations.

## 4. Conclusions

This paper presented a multiobjective optimization of a pedestrian bridge of steel-concrete composite structure, considering the cost, the emissions of carbon dioxide, and the comfort of the pedestrians. The set of Pareto-optimal solutions was obtained through the implementation in Python of a multiobjective Harmony Search algorithm, which uses non-dominated ranking and crowding distance metrics to classify the solutions. The program developed also performs the structural analysis of the structure and verifies the ultimate and serviceability limit states of the composite beam and concrete slab, according to the respective Brazilian codes. Two scenarios were studied, considering different values for the unit CO_2_ emission of the concrete. The first (A) uses unit emissions of a cradle-to-gate LCA from the on-site survey for the study region, while the second (B) are values obtained from the SimaPro software. The program developed was able to successfully obtain a Pareto front of non-dominated and feasible solutions, considering the three distinct objectives. With the analysis of the results, several conclusions are drawn which can be of aid during the decision-making in the design of sustainable pedestrian bridges of composite structure, taking into account the user’s comfort.

The results show that, for both scenarios, it is possible to significantly increase the comfort of pedestrians by a slight increase in the structure cost and emission. A structure can have its vertical acceleration reduced from 2.5 m/s^2^ to 1.5 m/s^2^ by an increase of 7% in its cost and to 1.0 m/s^2^ by an increase of 15%. For Scenario (A), an efficient way to reduce the vertical acceleration down to 1.0 m/s^2^ is to increase the web height of the I-beam, although further minimization must also increase superior and inferior flange width, slab thickness, and concrete strength. Scenario (B) showed a preference for changing the slab thickness, concrete strength, and flanges width, with solutions maintaining similar web heights while the acceleration is still reduced. For both cases, the optimal ratio for the web height and the total span (*L_e_*) is from *L_e_*/20 and *L_e_*/16. For Scenario (A), the web height grows from 850 mm while the vertical acceleration decreases until 1.0 m/s^2^, and from then on, the values stabilize at around 1050 mm. The optimal web heights for Scenario (B) are concentrated in both extremes in an almost constant behavior along vertical acceleration, with no values in the middle of the range.

In terms of costs, the steel I-beam is the main contributor to the total budget in both scenarios, with a 95% share of the value. On the other hand, the contribution of the concrete slab in the total emissions was raised from 26.42% to 45.57% in scenarios (A) to (B), given the bigger concrete unit emissions considered in the last scenario. In the first scenario, the costs and emissions grow together with the increase of the web height, trading off for lower vertical accelerations. However, in scenario (B), solutions with bigger web heights, minimum concrete strength, and slab thickness perform better in terms of CO_2_ emissions and vertical acceleration but with higher monetary costs. The web height is also important for the structure sizing since it is directly related to the displacements, the active constraint in most solutions. Nevertheless, the width of superior and inferior flanges is crucial to satisfy restrictions regarding the slenderness ratio and moments acting in the weak axis of the I-beam section. These results indicate that other design variables other than web height can play an important role in the vertical acceleration and sizing of the structure, evidencing the importance of them being considered in the decision-making process by engineers.

The behavior of the Pareto-optimal solutions suffered a considerable change for each scenario when analyzing the CO_2_ emissions and the cost of the pedestrian bridge. For the first scenario, the solutions presented a clear tendency of linear growth between the cost and emission, while in the second, the Pareto-optimal solutions were more dispersed. The optimal values of the design variables were also affected, with scenario (B) presenting a bigger consumption of concrete when compared to scenario (A). Therefore, two broader conclusions can be drawn from the variations in the results of the sensitivity analysis. First, the importance of considering different scenarios in optimization problems, as the results can be highly sensitive to changes in certain parameters. Second, the necessity of guaranteeing the reliability of the life cycle assessment is to make sure the results will be representative and applicable in practical ways, especially when raising sizing parameters for project development.

In spite of the fact that Brazilian Standards were considered in the computational implementation of the formulation and subsequent optimization, this aspect does not influence the results or the conclusions obtained once the same considerations and hypostasis were adopted in all the structures analyzed.

## Figures and Tables

**Figure 1 ijerph-20-03190-f001:**
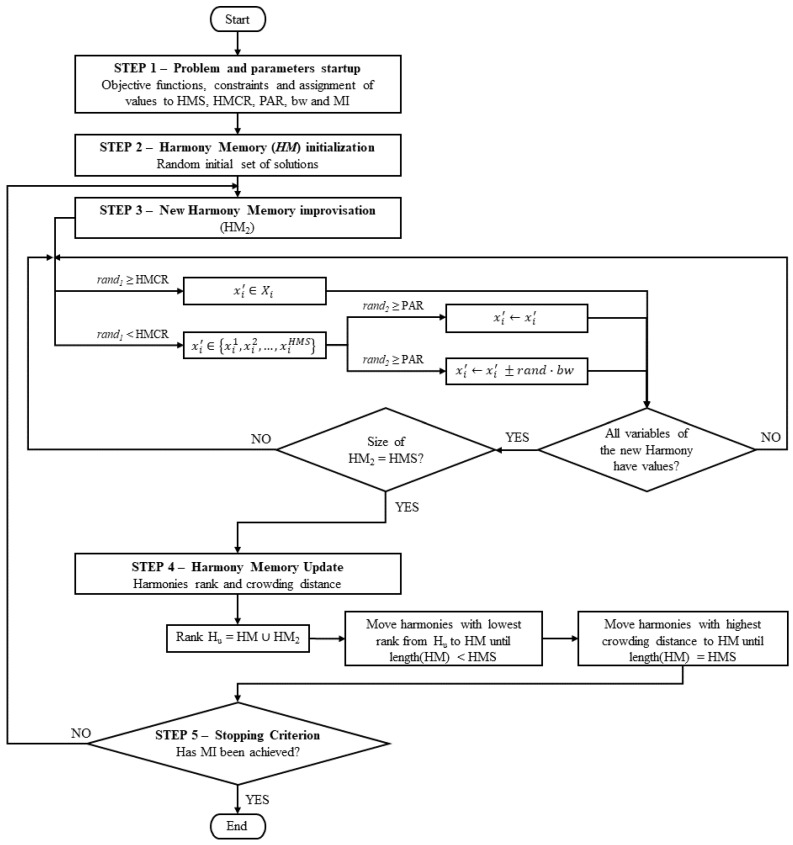
Multiobjective Harmony Search algorithm.

**Figure 2 ijerph-20-03190-f002:**
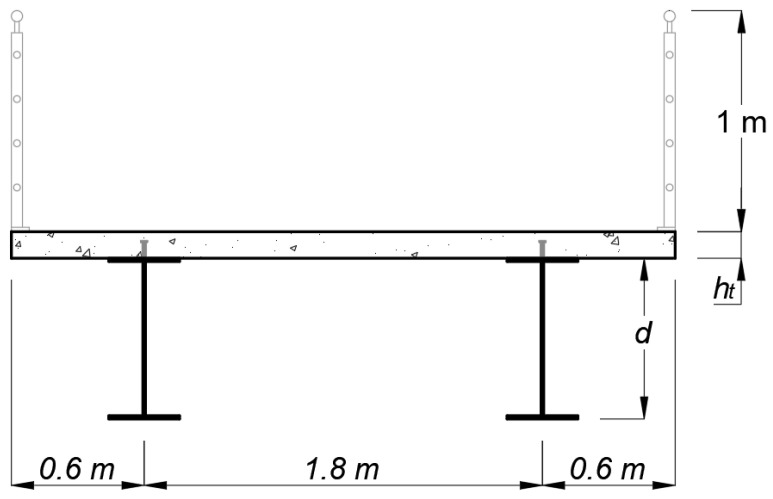
Pedestrian bridge cross-section.

**Figure 3 ijerph-20-03190-f003:**
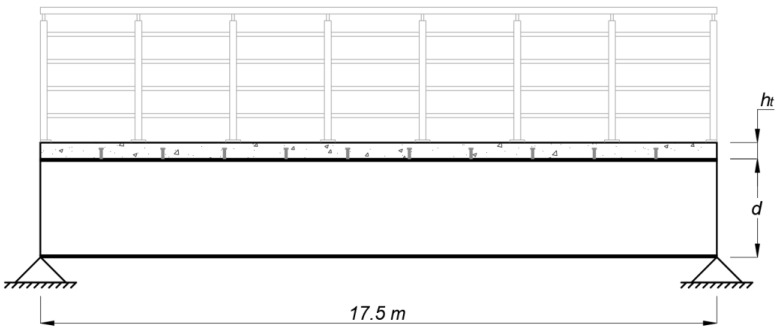
Longitudinal section of the pedestrian bridge.

**Figure 4 ijerph-20-03190-f004:**
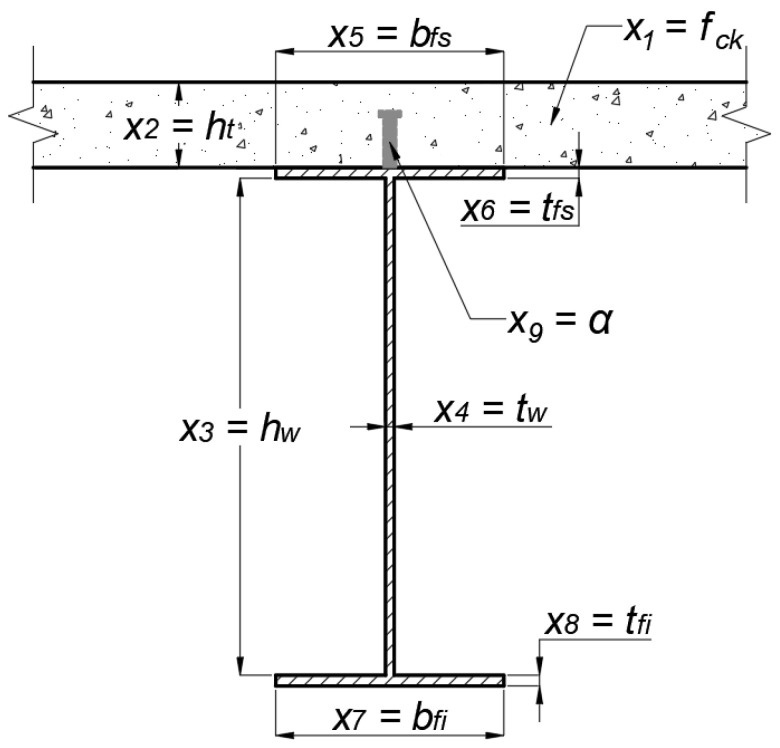
Variables of the optimization problem.

**Figure 5 ijerph-20-03190-f005:**
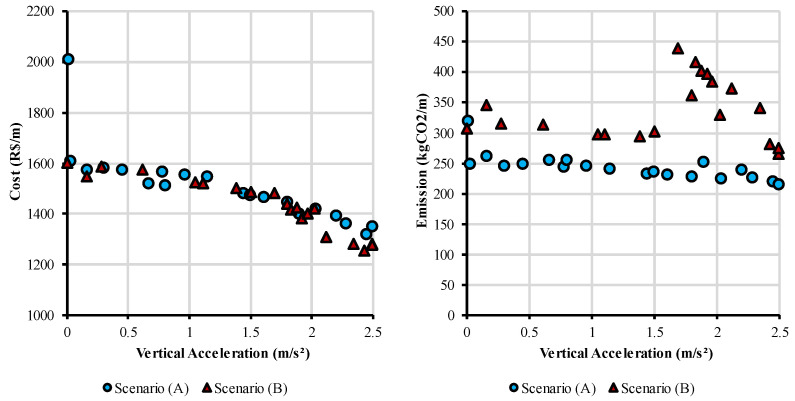
Pareto front for vertical acceleration versus cost and emission.

**Figure 6 ijerph-20-03190-f006:**
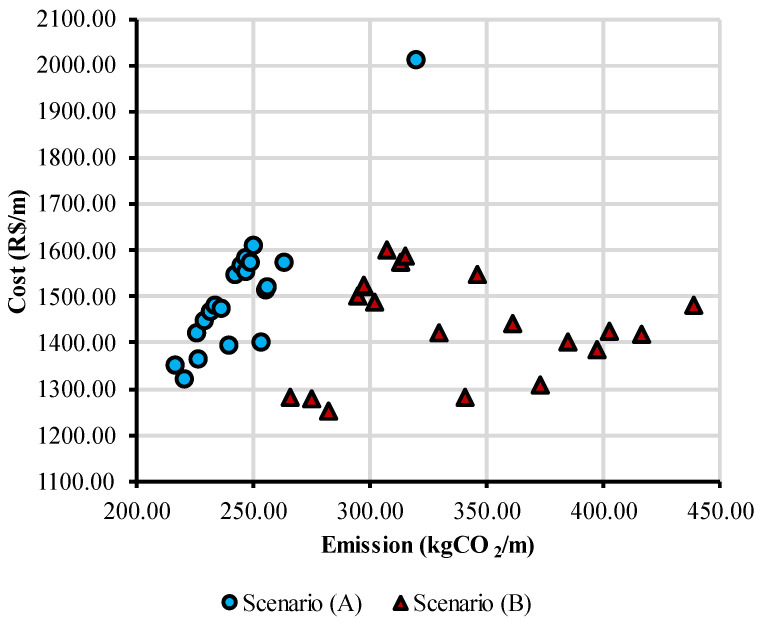
Pareto front for emissions and cost of the structure.

**Figure 7 ijerph-20-03190-f007:**
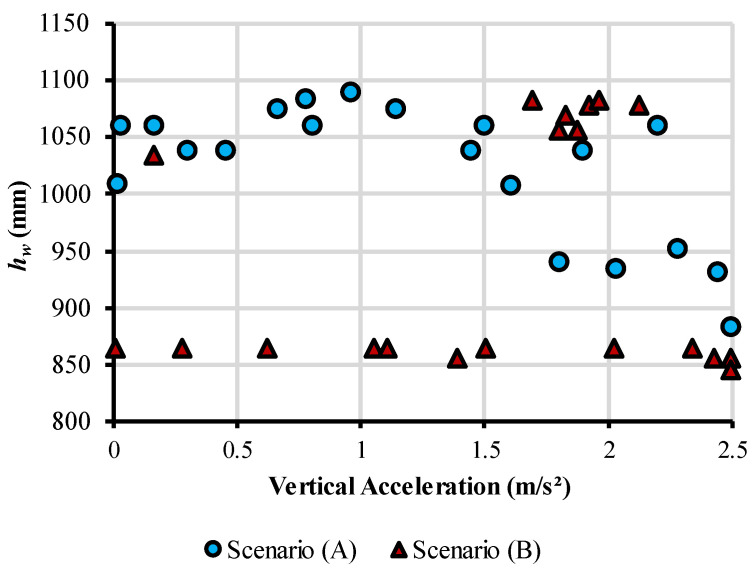
Web depth of I-beam versus vertical acceleration of pedestrian bridge.

**Figure 8 ijerph-20-03190-f008:**
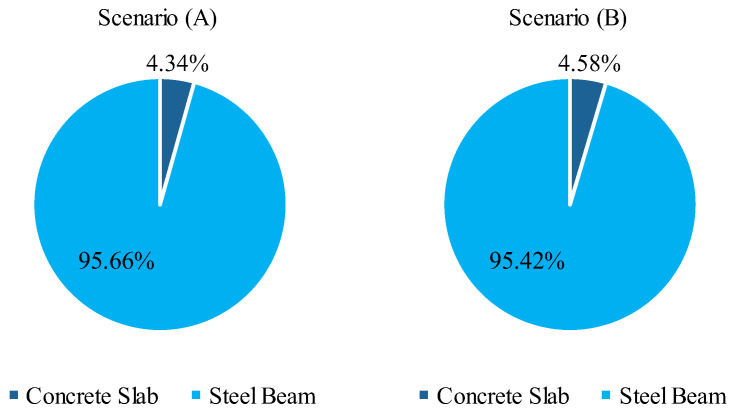
Contribution to the cost of each structural element.

**Figure 9 ijerph-20-03190-f009:**
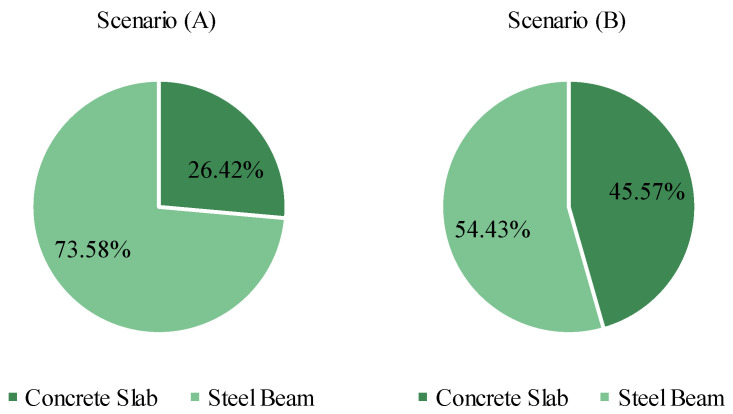
Contribution in emissions of each structural element.

**Figure 10 ijerph-20-03190-f010:**
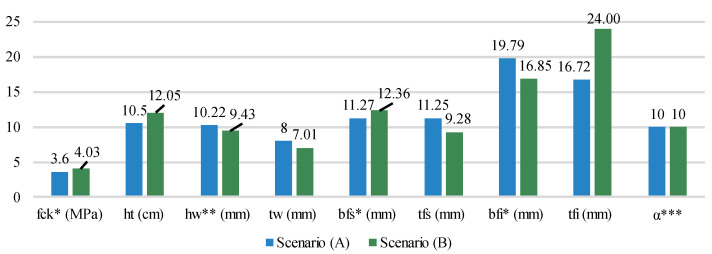
Average optimal values for design variables in each scenario * Values in the *y* axis divided by 10 ** Values in the *y* axis divided by 100 *** Values in the *y* axis multiplied by 10.

**Table 1 ijerph-20-03190-t001:** Values for multiobjective harmony search parameters.

Parameter	Value
HMS	20
HMCR	0.5
PAR_min_	0.1
PAR_max_	0.9
bw_min_	0.1
bw_max_	0.5
MI	500,000

**Table 2 ijerph-20-03190-t002:** Unit costs and emissions of each material.

Material	Unit	Cost (R$)	Emission A ^1^ (kgCO_2_)	Emission B ^2^ (kgCO_2_)
Concrete 30 MPa	m^3^	533.88	157.65	348.76
Concrete 45 MPa	m^3^	591.15	194.70	381.72
Concrete 50 MPa	m^3^	631.60	225.78	508.63
Reinforcement	kg	9.68	1.05	2.10
Steel I-beam	kg	14.56	1.91	1.91

^1^ Values used in scenario A optimization, where concrete and reinforcement emissions are from the LCA performed by Santoro and Kripka [27] to a city in the south of Brazil. ^2^ Values used in scenario B optimization, with emissions evaluated by Santoro and Kripka [18] using the SimaPro software, with adjustments in processes and quantities to match the same region.

## Data Availability

The data presented in this study are available on request from the corresponding author.

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
