# Peer review of "Multi-Objective Optimization Applied to the Design of Sustainable Pedestrian Bridges"

_ijerph, 2023, doi:10.3390/ijerph20043190_

Round 1

Reviewer 1 Report

This paper presented a multi-objective optimization of a pedestrian bridge of steel-concrete composite structure, considering the cost, the emissions of dioxide carbon, and the comfort of the pedestrians. The set of Pareto-optimal solutions was obtained through the implementation in Python of a multi-objective Harmony Search algorithm, which uses non-dominated ranking and crowding distance metrics to classify the solutions. The program developed also performs the structural analysis of the structure and verifies the ultimate and serviceability limit states of the composite beam and concrete slab, according to the respective Brazilian codes. Two scenarios were studied, considering different values for the unit CO2 emission of the concrete. Some revisions and explanations are needed before the consideration of publishing this paper.

1. A penalty factor was adopted to get the penalization in this paper, why the adopted value of the penalty factor is taken as 10000?

2. As it can be seen from the optimization results for emission and cost, a more dispersed result is observed for the solutions of Scenario (B). Please explain why these solutions are so dispersed?

3. The authors stated that the emissions in Scenario (B) showed a certain tendency to grow together with the costs. However, this conclusion is not so evident in Figure 6.

4. A case study of a specific pedestrian bridge of steel-concrete composite structure was conducted in this paper, so can the analysis method be applied to other forms of structures?

5. The authors should clarify what progress they have made based on the existing method in the literature.

Author Response

This paper presented a multi-objective optimization of a pedestrian bridge of steel-concrete composite structure, considering the cost, the emissions of dioxide carbon, and the comfort of the pedestrians. The set of Pareto-optimal solutions was obtained through the implementation in Python of a multi-objective Harmony Search algorithm, which uses non-dominated ranking and crowding distance metrics to classify the solutions. The program developed also performs the structural analysis of the structure and verifies the ultimate and serviceability limit states of the composite beam and concrete slab, according to the respective Brazilian codes. Two scenarios were studied, considering different values for the unit CO2 emission of the concrete. Some revisions and explanations are needed before the consideration of publishing this paper.

We are very grateful for the reviews, suggestions, and contributions to our paper entitled “Multi-objective Optimization Applied to the Design of Sustainable Pedestrian Bridges”. Based on them, several improvements were made to the text. The authors’ point-by-point comments on the revisions and responses are presented below.

  1. A penalty factor was adopted to get the penalization in this paper, why the adopted value of the penalty factor is taken as 10000?

A brief explanation of determining the penalty factor value has been added to the text, which now reads:

“In this paper, the penalization is obtained by the sum of the products of the constraint by a penalty factor. This value was determined based on a series of previous tests, where the lowest penalty factor that resulted in feasible solutions after the optimization end was adopted, with a value of 104 in this research.”

  1. As it can be seen from the optimization results for emission and cost, a more dispersed result is observed for the solutions of Scenario (B). Please explain why these solutions are so dispersed?

Thank you for the observation. A more in-depth discussion regarding the dispersion of results for Scenario (B) was included. The following paragraph was added to the text:

“This behavior for Scenario (B) can be mainly explained by a difference in the Pareto-optimal cross-section configuration. Some solutions showed a preference for bigger web heights, and smaller slab thickness and concrete strength, being the ones with the best performance in terms of pedestrian comfort and CO2 emissions. In contrast, other solutions adopted smaller web heights, bigger slab thickness, and higher concrete strengths, reducing the costs but trading off a poorer performance in terms of environmental impacts.”

  1. The authors stated that the emissions in Scenario (B) showed a certain tendency to grow together with the costs. However, this conclusion is not so evident in Figure 6.

This statement was rewritten for better understandability. Text now reads:

“On the other hand, in Scenario (B) the solutions are more dispersed, without a clear tendency between the two objectives. However, the cost and the CO2 emissions of the pedestrian bridge are not conflicting objectives, with solutions that are at the same time efficient in terms of costs and in environmental impacts.”

  1. A case study of a specific pedestrian bridge of steel-concrete composite structure was conducted in this paper, so can the analysis method be applied to other forms of structures?

The authors consider that it is possible to apply the analysis method and the developed program to other types of structures, given that certain changes in the code are made, such as in the structural analysis and in the pertinent verifications. In this sense, the following paragraph about this has been included in the text:

“Finally, the program developed in this research proved to be capable of obtaining a Pareto front of non-dominated solutions, which can aid in the decision-making process during the design of a steel-concrete composite pedestrian bridge. In addition, the MOHS algorithm proved to be applicable in the optimization of three distinct objectives for the structural engineering problem studied. It is worth mentioning that both the methodology of this work and the developed program are adaptable for different types of structures, given that certain changes in the code are made, such as in the structural analysis and in the pertinent verifications. This allows the study of other structural problems focused on their sustainability, and the efficiency of the optimization algorithm in different situations.”

  1. The authors should clarify what progress they have made based on the existing method in the literature

The progress in the literature achieved in this paper, mainly related to the optimization of vertical acceleration, is now better explored in the conclusions. The text now reads:

“The program developed was able to successfully obtain a Pareto front of non-dominated and feasible solutions, considering the three distinct objectives. With the analysis of the results, several conclusions are drawn which can be of aid during the decision making in the design of sustainable pedestrian bridges of composite structure, taking into account the user’s comfort.”

“A structure can have its vertical acceleration reduced from 2.5 m/s² to 1.5 m/s² by an increase of 7% in its cost, and to 1.0 m/s² by an increase of 15%. For Scenario (A), an efficient way to reduce the vertical acceleration down to 1.0 m/s² is to increase the web height of the I-beam, although further minimization must also increase superior and inferior flange width, slab thickness, and concrete strength. Scenario (B) showed a preference for changing the slab thickness, concrete strength, and flanges width, with solutions maintaining similar web heights while the acceleration is still reduced. For both cases, the optimal ratio for the web height and the total span (Le) is from Le/20 and Le/16. For Scenario (A), the web height grows from 850 mm while the vertical acceleration decreases until 1.0 m/s², and from then on, the values stabilize at around 1050 mm. The optimal web heights for Scenario (B) are concentrated in both extremes in an almost constant behavior along vertical acceleration, with no values in the middle of the range.

In terms of costs, the steel I-beam is the main contributor to the total budget in both scenarios, with a 95% share of the value. On the other hand, the contribution of the concrete slab in the total emissions was raised from 26.42% to 45,57% in scenarios (A) to (B), given the bigger concrete unit emissions considered in the last scenario. In the first scenario, the costs and emissions grow together with the increase of the web height, trading off for lower vertical accelerations. However, in scenario (B), solutions with bigger web heights, minimum concrete strength, and slab thickness perform better in terms of CO2 emissions and vertical acceleration, but with higher monetary costs. The web height is also important for the structure sizing since it is directly related to the displacements, the active constraint in most solutions. Nevertheless, the width of superior and inferior flanges is crucial to satisfy restrictions regarding the slenderness ratio and moments acting in the weak axis of the I-beam section. These results indicate that other design variables other than web height can play an important role in the vertical acceleration and sizing of the structure, evidencing the importance of them being considered in the decision-making process by engineers.”

The following text was also added to the abstract:

“For both scenarios, the optimal ratio for the web height and total span (Le) lies between Le/20 and Le/16. The web height, the concrete strength, and the slab thickness were the design variables with more influence in the value of the vertical acceleration.”

Reviewer 2 Report

This manuscript studied the issue of “Multi-objective Optimization Applied to the Design of Sustainable Pedestrian Bridges”.

First of all, I would like to thank the authors of this manuscript for the effort they put into making it. The paper needs to be rewritten and its objectives well redefined. On the other hand, I have added some comments with the main objective of improving the manuscript. In my opinion, the subject of the paper is remarkably interesting. However, the paper needs some major revisions.

i. What is the innovation point or significance of the study for this article? Please make clear the novelty and contribution of the manuscript and its results as compared to the extensive literature available. The paper does not provide a clear objective of the study.

ii. Results are merely present and there are no scientific findings are discussed. What is the difference between this manuscript and the article doi.org/10.1631/jzus.A1100304 or doi.org/10.1016/j.jclepro.2018.08.177 that was published before with related authors?

iii. In the paper, standard specification NBR 6123-88 and other is cited that can be replaced by new version such as ASTM and ACI, Moreover, the samples can satisfy these new standards.

iv. The introduction on similar work is very limited and does not cover similar experiences on the topic. I strongly recommend authors give a broader overview of similar works on the topic. The introduction should focus on the content related to the topic of the article.

v. In the selected value “slenderness ratio of steel beam and buckling analysis”, is not checked.

vi. Shear mechanism of weak plane in sustainable bridges are not mentioned.

vii. Material properties and behavior of steel I beams are not mentioned.

viii. What is the interaction between I beam section and concrete, (stud and the cost)?

ix. In conclusions, the useful data is not provided and the words prove general and lack of academic contributions. Please provide more specific details in the conclusions. The current conclusion is rather generic. The discussion is rather basic and short.

x. Some latest relative papers are not included. Such as:

doi.org/10.12989/cac.2022.29.6.407

doi.org/10.1016/j.engstruct.2022.114067

doi.org/10.12989/gae.2017.13.3.461

xi. The English language could be improved as well as the format of the manuscript; though overall English is acceptable.

Author Response

This manuscript studied the issue of “Multi-objective Optimization Applied to the Design of Sustainable Pedestrian Bridges”.

First of all, I would like to thank the authors of this manuscript for the effort they put into making it. The paper needs to be rewritten and its objectives well redefined. On the other hand, I have added some comments with the main objective of improving the manuscript. In my opinion, the subject of the paper is remarkably interesting. However, the paper needs some major revisions.

We are very grateful for the reviews, suggestions, and contributions to our paper entitled “Multi-objective Optimization Applied to the Design of Sustainable Pedestrian Bridges”. Based on them, several improvements were made to the text. The authors’ point-by-point comments on the revisions and responses are presented below.

  1. What is the innovation point or significance of the study for this article? Please make clear the novelty and contribution of the manuscript and its results as compared to the extensive literature available. The paper does not provide a clear objective of the study.

Thank you for the observation. The contribution of the paper is now better evidenced. In the conclusion, the text now reads:

“The program developed was able to successfully obtain a Pareto front of non-dominated and feasible solutions, considering the three distinct objectives. With the analysis of the results, several conclusions are drawn which can be of aid during the decision making in the design of sustainable pedestrian bridges of composite structure, taking into account the user’s comfort.”

“A structure can have its vertical acceleration reduced from 2.5 m/s² to 1.5 m/s² by an increase of 7% in its cost, and to 1.0 m/s² by an increase of 15%. For Scenario (A), an efficient way to reduce the vertical acceleration down to 1.0 m/s² is to increase the web height of the I-beam, although further minimization must also increase superior and inferior flange width, slab thickness, and concrete strength. Scenario (B) showed a preference for changing the slab thickness, concrete strength, and flanges width, with solutions maintaining similar web heights while the acceleration is still reduced. For both cases, the optimal ratio for the web height and the total span (Le) is from Le/20 and Le/16. For Scenario (A), the web height grows from 850 mm while the vertical acceleration decreases until 1.0 m/s², and from then on, the values stabilize at around 1050 mm. The optimal web heights for Scenario (B) are concentrated in both extremes in an almost constant behavior along vertical acceleration, with no values in the middle of the range.

In terms of costs, the steel I-beam is the main contributor to the total budget in both scenarios, with a 95% share of the value. On the other hand, the contribution of the concrete slab in the total emissions was raised from 26.42% to 45,57% in scenarios (A) to (B), given the bigger concrete unit emissions considered in the last scenario. In the first scenario, the costs and emissions grow together with the increase of the web height, trading off for lower vertical accelerations. However, in scenario (B), solutions with bigger web heights, minimum concrete strength, and slab thickness perform better in terms of CO2 emissions and vertical acceleration, but with higher monetary costs. The web height is also important for the structure sizing since it is directly related to the displacements, the active constraint in most solutions. Nevertheless, the width of superior and inferior flanges is crucial to satisfy restrictions regarding the slenderness ratio and moments acting in the weak axis of the I-beam section. These results indicate that other design variables other than web height can play an important role in the vertical acceleration and sizing of the structure, evidencing the importance of them being considered in the decision-making process by engineers.”

The objective was rewritten as well, and now reads:

“In this research, in addition to focusing on the sustainable design of pedestrian bridges, pedestrian comfort is jointly evaluated in terms of vertical accelerations caused by human-induced vibrations. The objective of this paper is to provide subsidies for decision making in the sustainable design of a pedestrian bridge with steel-concrete composite structure, aiming to minimize the vertical accelerations, as well as the cost, and the emission of carbon dioxide.”

  1. Results are merely present and there are no scientific findings are discussed. What is the difference between this manuscript and the article doi.org/10.1631/jzus.A1100304 or doi.org/10.1016/j.jclepro.2018.08.177 that was published before with related authors?

In contrast to the cited publications, this research focus on the multiobjective optimization of pedestrian bridges, performing verifications regarding the steel-concrete beams and the concrete slab. Beyond that, an important adding is the consideration of vertical acceleration as one of the objectives to be minimized, together with the cost and CO2 emissions.

Further discussions were added to the text, exploring deeper the findings of this paper. The following findings and discussions are included:

 “This behavior for Scenario (B) can be mainly explained by a difference in the Pareto-optimal cross-section configuration. Some solutions showed a preference for bigger web heights, and smaller slab thickness and concrete strength, being the ones with the best performance in terms of pedestrian comfort and CO2 emissions. In contrast, other solutions adopted smaller web heights, bigger slab thickness, and higher concrete strengths, reducing the costs but trading off a poorer performance in terms of environmental impacts.”

“The web height (hw) is a design variable that notoriously influenced the cost, the environmental impact, and the vertical acceleration generated by human-induced vibrations. This occurs due to the influence of the variable in the structure’s resistance and stiffness. For the Scenario (A), Pareto-optimal solutions with smaller hw have lower cost and environmental impact but perform poorer in comfort for the pedestrians, as Figure 7 shows. For the problem studied in this paper, the optimal values to hw, for Scenario (A), are in the range of 900 mm and 1100 mm. The figure also shows that, increasing this variable is an effective way to reduce the vertical accelerations from 2.5 m/s² to 1.0 m/s². However, to reduce accelerations further, other variables should be considered, such as the superior and inferior flange width, slab thickness, and concrete strength. The change in the behavior occurs as an attempt to satisfy the constraints regarding slenderness and the bending moment about the weak axis of the I-beam.

Figure 7. Web depth of I-beam versus vertical acceleration of pedestrian bridge.

Again, Scenario (B) presented a quite different behavior, although the interval of optimal values for hw is very similar. Two groups of Pareto-optimal solution stand out in Figure 7. The first remains practically constant around the value of 850 mm, while the second is grouped near the value of 1050 mm. For this scenario, variables such as slab thickness, concrete strength and superior and inferior flange width played a bigger role in obtaining solutions with lower vertical accelerations. The solutions with hw close to 850 mm have, in general, a slab thickness bigger than 14 cm, and a preference for concrete strength of 50 MPa. The other configurations, with bigger hw, have the minimal slab thickness and concrete strength. For both cases, the optimal ratio for hw and total span (Le) lies between Le/20 and Le/16.

Regarding the constraints of the optimization problem, the one that govern the structure sizing are, in the vast majority, the displacement, the I-beam slenderness and the horizontal bending moment that acts in the weak axis of the steel beam. As the displacement is the active restriction in almost every solution, the algorithm shows a preference for adopting the biggest hw that comply with the maximum slenderness, given that this is an effective way to increase the beam’s stiffness. However, solutions that increase only the slenderness ratio can have lower resisting capacity in the weak axis of the I-beam profile, and because of that, the constraint related to the verification of horizontal loads is also important in the sizing of the beam. For this restriction be satisfied, solutions must also increase the flanges width and thickness of the I beam cross-section.”

“Finally, the program developed in this research proved to be capable of obtaining a Pareto front of non-dominated solutions, which can aid in the decision-making process during the design of a steel-concrete composite pedestrian bridge. In addition, the MOHS algorithm proved to be applicable in the optimization of three distinct objectives for the structural engineering problem studied. It is worth mentioning that both the methodology of this work and the developed program are adaptable for different types of structures, given that certain changes in the code are made, such as in the structural analysis and in the pertinent verifications. This allows the study of other structural problems focused on their sustainability, and the efficiency of the optimization algorithm in different situations.”

  1. In the paper, standard specification NBR 6123-88 and other is cited that can be replaced by new version such as ASTM and ACI, Moreover, the samples can satisfy these new standards.

The reviewer is right about the importance of the consideration of standards such as ASTM and ACI. On the other hand, in the authors’ opinion, the use of Brazilian standards doesn’t invalidate the methodological procedure and the conclusions, once the comparisons and optimization were made on the same basis. In addition, it is important to notice that the parameters of international standards are similar to those applied in this study.

A new paragraph was added to the text to clarify the mentioned aspect. Text now sounds (in Conclusions):

In spite of the fact that Brazilian Standards were considered in the computational implementation of the formulation and in subsequent optimization, this aspect does not influence the results or the conclusions obtained, once the same considerations and hypostasis were adopted in all the structures analyzed.”

  1. The introduction on similar work is very limited and does not cover similar experiences on the topic. I strongly recommend authors give a broader overview of similar works on the topic. The introduction should focus on the content related to the topic of the article.

Thank you for this important observation. New related publications were added to the literature review. Text now reads:

“Researchers have applied optimization techniques in the design of pedestrian bridges with distinct objectives. Ferenc and Mikulski [19] performed a parametric optimization of a glass fiber reinforced polymer aiming to minimize its self-weight. Ferreira and Simões [20] minimized the cost of a cable-stayed pedestrian bridge with control devices. Penadés-Plà, García-Segura and Yepes [21] used a metamodel aided optimization to minimize the embodied energy of a concrete box-girder pedestrian bridge. In another research, Ferreira and Simões [22] optimized a cable-stayed steel pedestrian bridge with viscous dampers, with the objective of reducing the costs while satisfying dynamic verifications. The sustainability of a concrete box-girder pedestrian bridge is optimized in the publication of García-Segura et al. [23], minimizing the cost and CO2 emissions. The optimization considering tuned mass dampers in pedestrian bridges is a topic of interest in different publications [24-26]. Although several studies are developed regarding the optimization of costs and environmental impacts or the dynamic response of the pedestrian bridge, there is a lack of publications that consider the three objectives simultaneously. This justifies the research on the assessment of the interaction between the sustainability of the pedestrian bridge and its vertical acceleration generated by human-induced vibrations in a multiobjective optimization problem. Furthermore, the topic on optimization of steel-concrete composite pedestrian bridges is barely explored, as also pointed out by Yepes et al. [9].”

  1. In the selected value “slenderness ratio of steel beam and buckling analysis”, is not checked.

Both the slenderness ratio of the beams and the lateral buckling before the concrete hardening are verified in the optimization problem, and this is now better explained. The text now reads:

“The steel beams are also verified for the bending moments acting before the concrete hardening, and because the I section beams are unpropped during the structure construction, the lateral buckling in the construction phase is verified as well [43], considering a bracing length of 2.5 meters.”

  1. Shear mechanism of weak plane in sustainable bridges are not mentioned.

This mechanism is not mentioned in this paper because only the normative verifications that concern the studied problem were implemented, without additional considerations.

  1. Material properties and behavior of steel I beams are not mentioned.

This information is now mentioned, and the text reads:

“It is considered a steel ASTM A572 grade 50 for the I-beams, with 350 MPa of yield tensile strength and 200 GPa of modulus of elasticity.”

“The analysis considered a linear behavior of the steel I-beams and concrete slab.”

  1. What is the interaction between I beam section and concrete, (stud and the cost)?

A better description of the headed stud bolt was included in the text:

“The dimensions of the headed welding stud adopted are a nominal diameter of 19 mm and height of 135 mm.”

“The proposed optimization problem aims to minimize both the cost and the CO2 emissions associated with the construction phase of the pedestrian bridge. Equation (7) is used to evaluate the monetary cost of the structure (C), in function of the unit costs (ci) and the respective quantity of material consumed (qi), the elements (ec) considered and their unit is the volume of concrete, the mass of the slab’s reinforcement steel and the steel beam mass, which also considers the mass of the headed studs.”

  1. In conclusions, the useful data is not provided and the words prove general and lack of academic contributions. Please provide more specific details in the conclusions. The current conclusion is rather generic. The discussion is rather basic and short.

Several improvements were added both to conclusions and discussions. These changes are shown in revisions n.º 1 and n.º 2.

  1. Some latest relative papers are not included. Such as:

doi.org/10.12989/cac.2022.29.6.407

doi.org/10.1016/j.engstruct.2022.114067

doi.org/10.12989/gae.2017.13.3.461

We appreciate the indication of the papers. One of the publications most related to the scope of this research was included:

“Montoya, Hernández and Kareem [9] applied an aero-structural optimization to obtain optimal shape and size of a bridge deck.”

Furthermore, new related and recent publications were added to the text, as shown in revision n.º 4.

  1. The English language could be improved as well as the format of the manuscript; though overall English is acceptable

Some changes and several additions were made to the text.

Reviewer 3 Report

The paper describes the methodology and conclusions of a study conducted by the authors in which the multi-objective optimization of a steel-concrete composite pedestrian bridge is carried out. 

The objective of the study is to minimize cost, carbon dioxide emissions and vertical acceleration caused by human walking. 

The paper applies Multi-Objective Harmony Search (MOHS) to obtain non-dominated solutions and compose a Pareto Front. 

In the opinion of this reviewer, the paper is interesting, concise, clearly written and fits the scope of the journal. 

Therefore, this reviewer recommends its publication provided the following minor comments are properly addressed: 

1. In "Section 2.3.3.  Verifications and constraints" the beam erection process is not defined in the text. It is well known that, in composite beams, the erection process (e.g., whether or not the beams are propped during concrete pouring and hardening) is very relevant in the structural behaviour and safety verifications. It appears that in the study, the beams are not temporarily propped, but this should be briefly mentioned in the text for the sake of clarity. 

2. Line 223 describes that "The verifications of the ultimate limit state refer to the shear stress and the bending moment acting on the composite beam [34], before and after the concrete curing, as well as the bending moment on the concrete slab [35]." 

It should be clarified whether other failure modes have been considered. In particular, it is known that, in steel-concrete beams, lateral buckling of unrestrained steel beams during construction could govern the sizing of their top flanges. This reviewer wonders whether this could occur in the study shown in the article, especially since there is only one row of shear connectors welded to the center of the top flange.

3. In line 228-229 it is written "... the web thickness has to be bigger than the flanges thickness". This reviewer wonders if this is a typing error, since it is not a usual criterion. As far as I know, in steel-concrete beams, the flanges are usually thicker than the web, especially the bottom flanges, which are usually governed by bending moments. This should be carified.

4. Line 247. For the sake of clarity, it would be helpful if the reader could express the budget using a more widespread currency, such as the euro or U.S. dollar. In the opinion of this reviewer, at least the exchange rate (and its date) should be included. 

5. In line 274-5, there is a typo in the text. 

Author Response

The paper describes the methodology and conclusions of a study conducted by the authors in which the multi-objective optimization of a steel-concrete composite pedestrian bridge is carried out. The objective of the study is to minimize cost, carbon dioxide emissions and vertical acceleration caused by human walking. The paper applies Multi-Objective Harmony Search (MOHS) to obtain non-dominated solutions and compose a Pareto Front. In the opinion of this reviewer, the paper is interesting, concise, clearly written and fits the scope of the journal. Therefore, this reviewer recommends its publication provided the following minor comments are properly addressed:

The authors are very grateful for the reviews, suggestions, and contributions to our paper entitled “Multi-objective Optimization Applied to the Design of Sustainable Pedestrian Bridges”. Based on them, several improvements were made to the text. The authors’ point-by-point comments on the revisions and responses are presented below.

  1. In "Section 2.3.3. Verifications and constraints" the beam erection process is not defined in the text. It is well known that, in composite beams, the erection process (e.g., whether or not the beams are propped during concrete pouring and hardening) is very relevant in the structural behavior and safety verifications. It appears that in the study, the beams are not temporarily propped, but this should be briefly mentioned in the text for the sake of clarity.

Thank you for the observation. It is now mentioned in the texts that the composite beams are unpropped during the construction phase, as well as including the necessary security verifications in this situation. The text now reads:

“The pedestrian bridge is simply supported, with a single span of 17.5 m to be surpassed. During the construction phase, it was considered that the steel beams are unpropped.”

  1. Line 223 describes that "The verifications of the ultimate limit state refer to the shear stress and the bending moment acting on the composite beam [34], before and after the concrete curing, as well as the bending moment on the concrete slab [35]." It should be clarified whether other failure modes have been considered. In particular, it is known that, in steel-concrete beams, lateral buckling of unrestrained steel beams during construction could govern the sizing of their top flanges. This reviewer wonders whether this could occur in the study shown in the article, especially since there is only one row of shear connectors welded to the center of the top flange.

This part has been rewritten, explaining that the beam is also checked for the bending moment acting before de concrete hardening, including the lateral buckling. The text now reads:

“The verifications of the ultimate limit state refer to the shear stress and the bending moment acting on the composite beam [43], as well as the bending moment on the concrete slab [44]. The steel beams are also verified for the bending moments acting before the concrete hardening, and because the I section beams are unpropped during the structure construction, the lateral buckling in the construction phase is verified as well [43], considering a bracing length of 2.5 meters.”

  1. In line 228-229 it is written "... the web thickness has to be bigger than the flanges thickness". This reviewer wonders if this is a typing error, since it is not a usual criterion. As far as I know, in steel-concrete beams, the flanges are usually thicker than the web, especially the bottom flanges, which are usually governed by bending moments. This should be clarified.

This really was an unnoticed typing error, and it has been corrected to:

“The depth of the beam must be 50% bigger than the inferior flange width, and the flanges thickness must be bigger than the web thickness.”

  1. Line 247. For the sake of clarity, it would be helpful if the reader could express the budget using a more widespread currency, such as the euro or U.S. dollar. In the opinion of this reviewer, at least the exchange rate (and its date) should be included.

The exchange rate in relation to the US dollar is now included in the text, and conversions are made in the discussions for making it easier for the readers to understand the monetary values. The text now reads:

“These values are given in Table 2, where the monetary costs are expressed in real (R$), the Brazilian currency. At the date of January 22, 2023, the exchange rate against the US dollar is R$ 1.00 for $ 0.19 (or $ 1.00 for R$ 5.21).”

“In both scenarios, the costs of non-dominated solutions are very similar and stay within the range of R$ 1400.00 ($ 268.82) and R$ 1600.00 ($ 307.23) per meter of the pedestrian bridge, with a singular exception that costs approximately R$ 2000.00 ($ 384.03) per meter.”

  1. In line 274-5, there is a typo in the text.

The typo has been corrected to:

“On the other hand, the concrete slab shares a considerable part of the structure’s emissions, as illustrated in Figure 9, for the lowest emission 

Round 2

Reviewer 1 Report

The paper can be accepted for publication in its present form.

Reviewer 2 Report

The paper after minor revision can be accepted for publication.